# Characterization of Mandibular Border Movements and Mastication in Each Skeletal Class Using 3D Electromagnetic Articulography: A Preliminary Study

**DOI:** 10.3390/diagnostics13142405

**Published:** 2023-07-19

**Authors:** Nicole Constanza Farfán, María Florencia Lezcano, Pablo Eliseo Navarro-Cáceres, Héctor Paulo Sandoval-Vidal, Jordi Martinez-Gomis, Loreto Muñoz, Franco Marinelli, Ramón Fuentes

**Affiliations:** 1Research Center for Dental Sciences (CICO), Dental School, Faculty of Odontology, Universidad de La Frontera, Temuco 4811230, Chile; nicole.farfan@ufrontera.cl (N.C.F.); pablo.navarro@ufrontera.cl (P.E.N.-C.); franco.marinelli@ufrontera.cl (F.M.); 2Department of Integral Adults Dentistry, Dental School, Faculty of Odontology, Universidad de La Frontera, Temuco 4811230, Chile; 3Cybernetics Laboratory, Department of Bioengineering, Universidad Nacional de Entre Ríos, Entre Ríos CP E3100, Argentina; lezcano.f@gmail.com; 4Department of Pediatric Dentistry, Orthodontic Dental School, Faculty of Odontology, Universidad de La Frontera, Temuco 4811230, Chile; paulo.sandoval@ufrontera.cl; 5Department of Prosthodontics, School of Dentistry, Faculty of Medicine and Health Sciences, University of Barcelona, 08907 Barcelona, Catalonia, Spain; jmartinezgomis@ub.edu; 6Oral Health and Masticatory System Group, (Bellvitge Biomedical Research Institute) IDIBELL, L’Hospitalet de Llobregat, 08907 Barcelona, Catalonia, Spain; 7Carrera de Odontología, Facultad de Odontología, Universidad de La Frontera, Temuco 4811230, Chile; l.munoz12@ufromail.cl

**Keywords:** skeletal class, mandibular movement, mandibular border movement, mastication

## Abstract

Mandibular movement recording is relevant for the planning and evaluation of mandibular function. These movements can include mandibular border movements (MBM) or mastication. Our objective was to characterize the kinematics of MBM and mastication among skeletal classes I, II, and III in the three spatial planes. A descriptive cross-sectional study was conducted with 30 participants. Instructions were provided on how to form Posselt’s envelope and to perform masticatory. After data processing, we obtained numerical values for the areas, trajectories, and ranges of MBM that formed Posselt’s envelope and the values for speed, masticatory frequency, and the areas of each masticatory cycle. Significant differences were found in the area of Posselt’s envelope in the horizontal plane between skeletal classes I and III and in the range of right laterality between skeletal classes II and III. Mastication showed significant differences in the area of the masticatory cycles in the horizontal plane between classes I and III and between classes II and III. In conclusion, there were differences in MBM and mastication between skeletal classes III and I in the horizontal plane. This study supports the need to establish normal values for mandibular kinematics in skeletal class III.

## 1. Introduction

The method of recording mandibular movements (MM) was introduced in dentistry and mainly in oral physiology as a planning tool for diagnosing and evaluating rehabilitation treatments by analyzing the kinematics of the gliding movement of the mandible [1,2], such as speed, trajectories, and ranges [3,4,5]. These measurements are important for examining patients with possible functional disorders of the masticatory system [6]. MM are produced by the interaction of rotational and translational movements, which occur simultaneously and are coordinated by the temporomandibular joint (TMJ) [7]. These movements can be classified as mandibular border movements (MBM) and mandibular non-border movements. MBM occur when MM utilize the maximum gliding capacity of the TMJ and are limited by the morphology and ligaments of the TMJ. The recording of the upper edge movement and opening in three spatial dimensions (3D) comprises Posselt’s envelope [7]. Mandibular non-border movements, such as mastication, occur during mandibular function and are considered free movements. These movements are determined by conditional responses of the neuromuscular system [7].

In recent decades, technological improvements in the techniques of follow-up position have made it possible to record the dynamics of the joints with high temporal and spatial resolution [6]. The 3D assessment of MM is possible through 3D electromagnetic articulography (3D EMA), which can evaluate geometric variables (morphology) and kinematics (areas, trajectories, ranges, and speed) of the MBM and masticatory movements [4,5,6,7,8] and replaces the old 2D follow-up systems that used large face bows that made MM assessment difficult.

According to the World Health Organization (WHO), malocclusions are highly prevalent among oral health issues [9]. In the first half of the 20th century, cephalometry was described as a useful tool for diagnosing and evaluating malocclusions [10]; it is currently considered the best way to establish skeletal relationships [11]. The study of MM in subjects with skeletal malocclusions or alterations is very important for orthodontic treatment, since they are part of the functionality analyses of the stomatognathic system; for example, ranges and trajectories of MM are evaluated before and after orthognathic surgeries [12].

The evidence is limited when analyzing the kinematics of the MBM and masticatory movements in the different skeletal classes; it has been found that skeletal class III is the most thoroughly researched. Studies have focused mainly on the changes that MM produced after undergoing orthognathic surgical procedures and there has been little evidence of the characterization of these MM in the different skeletal classes [12,13,14]. Accordingly, there have been reports that MBM and masticatory movements changed after surgery [15,16], seeing a significant increase in the maximum mandibular excursions after orthognathic surgery [16], whereas other studies have described a postoperative reduction in the maximum opening (MO) and lateral movements. And, even though these values increased a year-and-a-half after surgery, they did not reach the preoperative values; there were also post-operative changes in the trajectory of mastication [17]. Changes in the mastication pattern have been reported in skeletal class III, passing from a linear pattern to a posterolateral pattern with the surgery [3].

Although there have been reports of the MBM and masticatory movements in the skeletal classes, these were conducted with low-precision instruments and were not focused on characterizing MBM and mastication patterns but rather on evaluating the functional changes produced after orthognathic surgery. Our research objective was to characterize the MBM and masticatory movements of skeletal classes I, II, and III through 3D electromagnetic articulography.

## 2. Materials and Methods

This descriptive cross-sectional study analyzed the MBM and masticatory movements in the three spatial planes (frontal, sagittal, and horizontal) of subjects of all skeletal classes (I, II, and III) who requested orthodontic treatment at the Dental School of the Universidad de La Frontera, Chile. The sample was selected by non-probability convenience sampling. A total of 30 participants were included in the study: 13 men and 17 women (23 ± 3.5 years of age). A total of 3 groups were formed: classes I, II, and III, with 10 participants each.

Inclusion criteria: subjects of both sexes with complete permanent dentition up to the second molar, requiring and requesting corrective orthodontic treatment in the Dental School of the Universidad de La Frontera, Chile.

Exclusion criteria: subjects who have undergone previous orthodontic treatment, absence of one or more teeth (not including a third molar), who have dental implants or dental prostheses, participants allergic to peanuts or with oral lesions that could prevent or limit the correct performance of the DPM, and those with signs or symptoms of TMJ disorders ruled out by application of a clinical examination and a self-reported test recommended by the American Academy of Orofacial Dolor [18].

Participants who met our selection criteria were attended to in the Oral Physiology Laboratory of the Centro de Investigación en Ciencias Odontológicas (Universidad de La Frontera, Temuco, Chile), where MBM and masticatory movements were recorded. This study was approved by the Scientific Ethics Committee of the Universidad de La Frontera (Approval number 078/2017). Per the World Medical Association’s Declaration of Helsinki (2008), written informed consent was obtained from volunteers before participation after they were informed about the nature of the study.

### 2.1. Data Collection

The participants who requested orthodontic treatment at the Universidad de La Frontera dental clinic had to be classified according to the skeletal class; for this, an imaging study was carried out with lateral teleradiography using the ANB angle analysis. This analysis was carried out by a single operator (the orthodontist in charge of diagnosis), who obtained the ANB angle measurement from the Steiner analysis, establishing the discrepancy in the sagittal direction between the maxilla and the mandible. This ANB angle was plotted by joining point A (the maximum concavity of the anterior border of the maxilla), point B (the maximum concavity of the mandible), and point N (the nasion). The angle obtained from these points had an expected value of 0° to 4°, which was defined as class I; a lower value (<0°) was defined as class III and a higher value (>4°) as class II [19].

Once the skeletal class was determined, the participants were invited to be evaluated whether they met the selection criteria for the present study in the oral physiology laboratory of the Universidad de La Frontera; those who were eligible underwent MBM and masticatory evaluations. A 3D electromagnetic articulograph (AG501; Carstens Medizinelektronik) was used to record mandibular border and functional movements. The recording protocol was based on that developed by Fuentes et al. [8,20,21] and Vargas-Agurto et al. [22]. The border movements recorded were based on those proposed by Okeson [7]. Each clinical examination and experimental procedure were carried out by the same researcher (N.F.B).

Four sensors were placed on the participant’s head (Figure 1A) at the mastoid skin points: right (1st) and left (2nd), glabella skin point (3rd), and mandibular interincisor (4th) (Figure 1B). The first three sensors were used as reference sensors in the head correction procedure that allowed the device to eliminate natural head movement and record only the movement of the mandibular sensor (4th). These sensors were attached to the subject with biologically compatible glue (Epiglu^®^, Meyer Haake, Germany) in all skeletal classes (Figure 2). Before starting the recording of MM, a 5 s baseline test was performed and the reference sensors were sampled with the participant facing forward and the Frankfurt plane parallel to the floor. With these data, the head correction was performed once, before starting the measurements. All movements started with the participant in maximum intercuspation position (MIP) and were performed three times with three minutes of rest between each movement.

The first movement to be recorded was the MO, starting from MIP and ending in MO.

To perform the necessary to form Posselt’s envelope in the frontal plane, participants performed two MMs according to the following instruction: move the jaw from MIP to maximum movement lateral with dental contact right (MLC-R) and from that point to MO with edge opening. To complete Posselt’s envelope, participants returned to MIP. The second movement was a maximum movement lateral with dental contact left (MLC-L). From that point, an edge opening to MO was performed (Figure 3A). Both movements were recorded. To perform the MBM necessary to form Posselt’s envelope in the sagittal plane, participants performed two MMs according to the following instruction: move the jaw from MIP until maximum retrusion position (MRP) is performed; from that point, perform a posterior bordering aperture to MO. To complete Posselt’s envelope, participants returned to MIP; the second movement was to the maximum protrusion position (MPP) and, from that point, an anterior edge MO was performed (Figure 3B). Both movements were recorded. To perform Posselt’s envelope in the horizontal plane, participants performed two MM according to the following instruction: move the jaw from MIP to MLC-R and, from that point, perform the movement to the maximum protrusion position (MPP). Then, to complete Posselt’s envelope, perform a return to MIP and MLC-L and, from that point, perform the movement to the MPP (Figure 3C). Both movements were recorded. The last MM recorded were the masticatory movements of 3.7 g of peanuts, according to previous reports in the literature [23,24]. The masticatory movement recording started with the participant in MIP and a peanut between the tongue and palate. The participant was then asked to chew freely without indicating any side or the number of masticatory cycles. The recording ended before initiating the first swallow.

### 2.2. Data Processing and Outcome Variables

For the calculations, the three recordings made in each MM were considered. All data were recorded, labeled, and transferred from the EMA 3D AG501 device to another computer for processing. The data were stored in binary files (.pos) containing each sensor’s position with a sampling rate of 250 Hz. They were processed with MATLAB software version 2018a (V2018a) (The Math Works, Inc., Natick, MA, USA) using the position data matrix in binary files (.pos). Specific calculation routines (scripts), developed especially for this study, were used to obtain the numerical values of the areas, trajectories, ranges, masticatory movement velocities, and masticatory frequencies.

Once the data were processed, the following variables were analyzed: MO, MBM, and masticatory movements. In the MO movement, trajectory and range were measured from MIP to MO. With the MBM data, Posselt’s envelope was generated in the frontal, sagittal, and horizontal planes; the kinematics were analyzed through the areas, trajectories, and ranges. The area of Posselt’s envelope was measured for each plane.

In the frontal plane, the trajectory was measured for the right and left laterality from MIP to the right or left maximum laterotrusive dental contact point (MLC). The right and left opening trajectories were also measured from the right or left maximum laterotrusive point to MO (Figure 3A). In the sagittal plane, the trajectory of the bordering posterior opening was measured, starting from MIP or the maximum retrusion position (MRP) and ending at MO. The trajectory of the frontal opening bordering was measured from MIP to the maximum protrusion position (MPP) and ending at MO. The range measured was the protrusion range (PR) from the MIP to the MPP (Figure 3B). In the horizontal plane, the trajectories were measured by starting the movement from MIP to the maximum laterality dental contact right and ending at the maximum protrusion position (MPP). This was repeated on the left side. The lateral range (RL) was measured from MIP to the MLC-R or MLC-L (Figure 3C). With the processing of the movement masticatory data, the area of the masticatory cycles (mm^2^) in the frontal, sagittal, and horizontal planes, the mandibular opening and closing speed (mm/s), and the masticatory frequency (cycles/minutes) were obtained.

### 2.3. Variables Analyzed

Area: both Posselt’s envelope and the masticatory cycles were calculated in mm^2^.

Trajectory: This was considered the gliding pathway of a 3D movement. This study corresponded to the route of the mandibular sensor, the coordinates on the *x, y*, and *z* axes. Equation (1) contains the coordinates of the points that comprised the trajectory (*T*) and the number of points of each recorded trajectory.

Range: this was calculated as a 2D linear distance between the beginning and end of the movement with two defined points.

Speed masticatory movements: the mandibular opening and closing movements were evaluated in mm/s.

Masticatory frequency: this was measured in cycles per minute.
(1)$$T=\sum_{i=1}^{n-1}\sqrt{{\left(x_{i}-x_{i+1}\right)}^{2}+{\left(y_{i}-y_{i+1}\right)}^{2}+{\left(z_{i}-z_{i+1}\right)}^{2}}$$

### 2.4. Statistical Analysis

Descriptive statistics (mean + SD) were used (Kolmogorov–Smirnov normality test (*p* > 0.05)). Gender differences in MBM and mastication were assessed by the Mann–Whitney U test and differences between angle class groups were assessed by the Kruskal–Wallis test (pairwise comparisons and significance values of s were adjusted by the Bonferroni correction for multiple testing). A statistical analysis was performed using the SPSS program (IBM SPSS, v. 2 7.0.1.0); the significance threshold was 5%.

## 3. Results

The analysis included all the recordings made for each participant and from each MM. No gender differences were detected with regard to the MBM or masticatory movements. In the MO, the trajectory and range showed no significant differences between the groups analyzed (Table 1).

In Posselt’s envelope in the frontal and sagittal planes, no significant differences were observed in the variables analyzed between groups (Table 2 and Table 3); however, in Posselt’s envelope in the horizontal plane, significant differences were found (*p* = 0.046, Kruskal–Wallis test) in the area between skeletal classes I (107.7 mm^2^) and III (58.8 mm^2^) and in the lateral range (*p* = 0.042, pairwise comparisons) between skeletal classes II (9.2 mm) and III (6.4 mm) (Table 4).

Mastication showed significant differences in the area of masticatory cycles in the horizontal plane between groups (*p* = 0.003, Kruskal–Wallis). Pairwise comparison showed significant differences between classes I (10.9 ± 8 mm^2^) and III (18.4 ± 6 mm^2^) (*p* = 0.036) and between class II (8.1 ± 3.9 mm^2^) and III (18.4 ± 6 mm^2^) (*p* = 0.004) (Table 5). Figure 4 is a graphical representation of the areas of the cycles in each plane (frontal, sagittal, and horizontal) for each skeletal class.

## 4. Discussion

This study provides new previously undescribed information on how a skeletal pattern influences mandibular function tests, such as kinematics MBM and masticatory movements, by applying a modern and accurate assessment method. Knowing this variability between skeletal classes is a useful tool for clinicians and researchers involved in treating mandibular anomalies as it is an indicator of mandibular function and is often assessed before and after corrective malocclusion treatment, such as orthognathic surgery [25].

Several methods, varying in their accuracy, have been described to assess mandibular function, from using rulers, clamps, and videos, which, although simple, quick to apply, and inexpensive, present questionable accuracy and systematization [26,27]. Regarding the devices used in skeletal classes, opto-electronic systems and opto-electronic gnatho-hexagraphs have been reported, all of which use a facebow, which is considered a disadvantage since its large dimensions can make it difficult for the participants to perform MM and may alter the recorded values. The gold standard is the ARCUSdigma computerized axiograph, which allows for qualitative analysis of mandibular movement; however, it also uses a facebow. In our study, a modern 3D EMA system was used that could record mandibular movement with high spatial (0.3 mm) and temporal (1 KHz sampling rate) accuracy in all three spatial planes [28], allowing assessment of MM geometry and kinematics using small sensors that did not interfere with mandibular displacement. This made it possible to analyze movements in a more complex manner and to provide a characterization of the movement with reliable and accurate data [29].

Improving the functionality of the stomatognathic system is one of the objectives of orthognathic surgery. Various surgical techniques seek to normalize variables of mandibular function, for which the evaluation of parameters such as speech, mastication, and MBM are necessary to determine the functional results of the surgical techniques used [30]. There are several studies, with varied information. Some report a reduction of the MBM amplitude as a complication of orthognathic mandibular advancement and retraction surgeries [3,31], as well as a significant change in the duration of the masticatory cycles and in the pattern of masticatory movement, going from a linear pattern with a long opening to a movement of greater laterality [15,25]. Others suggest that mandibular advancement surgery in skeletal class II subjects reduces mandibular mobility more than other orthognathic surgical procedures [32,33].

According to what has been published in the literature, skeletal class III is the most studied; research evaluates MBM and masticatory movements separately associated with the evaluation of treatments such as orthognathic surgeries [5,12,13,15,25]. Our results confirmed that skeletal class III presented lower MBM parameters than classes I and II, such as the area of the polyhedron in the horizontal plane and the right lateral range of motion; a greater area of the masticatory cycle in the horizontal plane was also demonstrated for skeletal class III compared with classes I and II. This justified the objective of restoring normal functional parameters of skeletal class III with adequate orthodontic treatment.

The mechanism that explained our results and the discrepancies between skeletal class III and classes I and II could include the skeletal and dental anatomy of skeletal class III, since, when starting the MM from a more advanced mandibular position, there was little space available to perform MBM in the horizontal plane. However, because of the altered overjet and overbite, there was no obstruction to the molars sliding in this plane during mastication. This could explain why skeletal class III had a larger masticatory area in the horizontal plane and a lower right range in the MBM than skeletal classes I and II. The functional and postural lateral differences described in several studies, some of which have been linked to masticatory asymmetry, can be used to explain the lateral differences seen in this study, where only the right lateral range was greater in skeletal class II [34]. Although masticatory function is not usually symmetrical [35], the reason why skeletal class was related to MBM rank only on the right side could be due to the significant values on the right (*p* = 0.035) and left (*p* = 0.065).

Not all the parameters evaluated in this study have been previously reported in the literature. The maximum buccal opening movement is one of the most studied, as well as the maximum right and left lateral movements; however, the literature analyzing the movements presents a low level of precision when specifying whether to record the trajectory or the range of the MM and the reference points considered for recording the MM. Ueki et al., 2020 [5] used the centric relation position as the start of the movements. This position has been highly questioned in the literature as it has been challenging to reach a consensus on its definition, so it tends to be poorly understood by dentists, given the wide variety of anatomical structures involved in the position and its deficiency in clinical reproduction [36].

The limitations of this study included the small sample size for the skeletal class groups. Overbite and overjet were not evaluated as they were not taken into account in the preliminary study design; however, these variables could help to understand the discrepancy in the results of this study. The subdivision of skeletal class II (division 1 and division 2) was also not considered; these parameters will be examined in future studies, as will the control of variable bias.

In light of the results, and despite the limitations, it should be noted that we have offered an adequate methodology to assess PMD and masticatory movements accurately and that the data collected in this study support the need to return normal values of the mandibular function to skeletal class III since part of their MBM and masticatory function are different from those of class I. This information could be used as a reference to evaluate mandibular function before and after treatment to correct malocclusions.

## 5. Conclusions

We conclude that there are differences in mandibular border and functional movements between skeletal classes III and I–II in the horizontal plane. Skeletal class III performs the lowest ranges of right lateral maximal movement and has a smaller area of Posselt’s envelope in this plane; however, it performs masticatory cycles of a larger area in that plane. This study supports the need to return normal values of mandibular kinematics to skeletal class III.

## Figures and Tables

**Figure 1 diagnostics-13-02405-f001:**
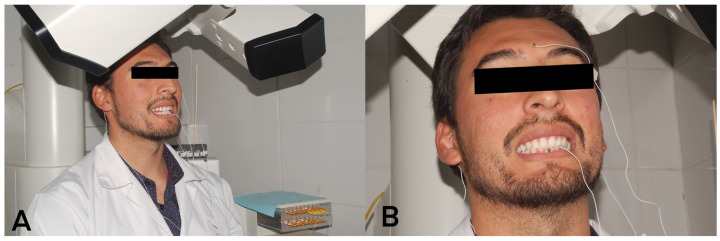
(**A**) Patient representation in AG501 3D-EMA, (**B**) EMA AG 501 sensor at glabella skin point and at the mandibular interincisor.

**Figure 2 diagnostics-13-02405-f002:**
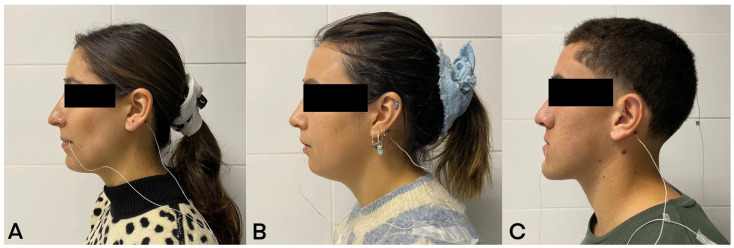
Sagittal plane, AG501 3D-EMA sensors in left mastoid skin point in participants. (**A**) Skeletal class I; (**B**) skeletal class II; (**C**) skeletal class III.

**Figure 3 diagnostics-13-02405-f003:**
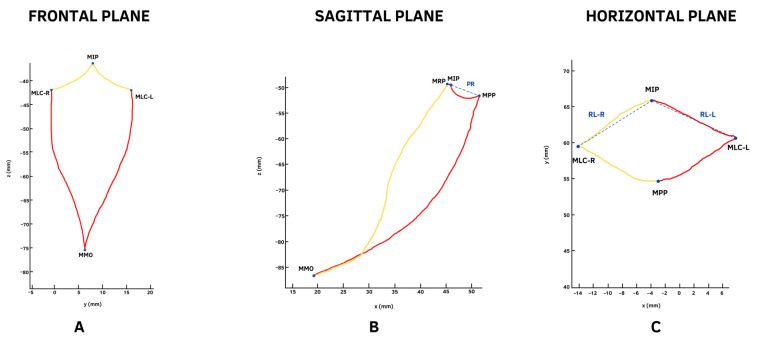
Posselt’s envelope in the frontal (**A**), sagittal (**B**), and horizontal (**C**) planes, formed by the union of the trajectories of the previously instructed MBM; the first movements performed are represented with yellow lines and those that completed Posselt’s envelope are represented with red lines. The dotted blue lines represent the linear range between two points: in the sagittal plane, the protrusion range (PR); in the horizontal plane, the right lateral range (RL-R) and left lateral range (RL-L).

**Figure 4 diagnostics-13-02405-f004:**
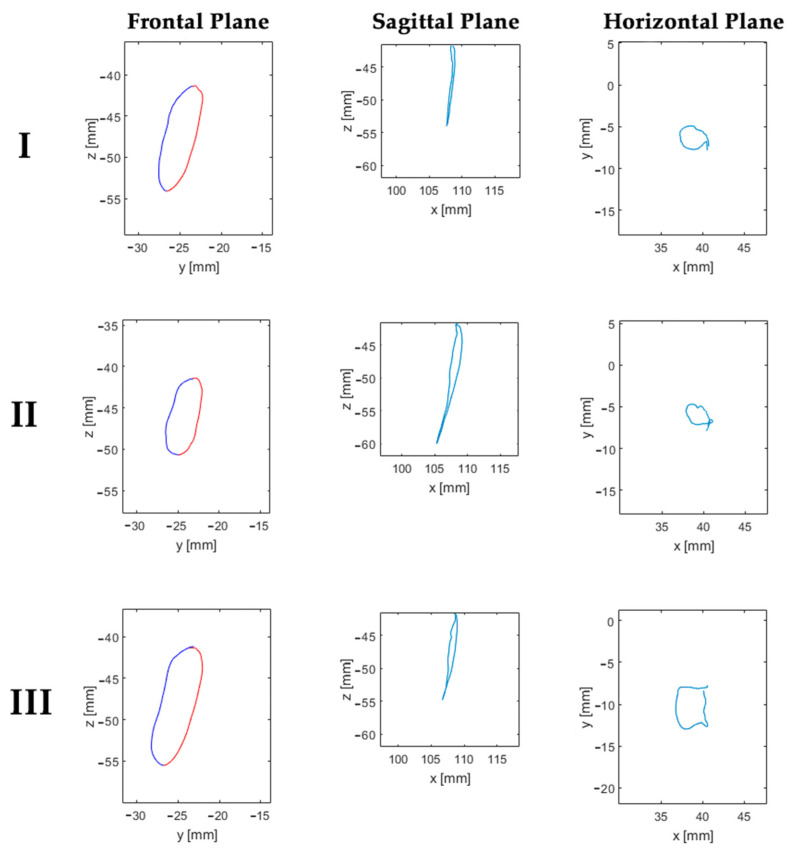
Example of masticatory cycle areas of each skeletal class in the frontal, sagittal, and horizontal planes. Masticatory cycles frontal plane: blue line upward trajectory, red line downward trajectory.

**Table 1 diagnostics-13-02405-t001:** Trajectory and range of maximum opening.

Skeletal Class	Maximum Opening (Mean ± SD, mm)	
	Trajectory	Range
I	65.1 ± 6.5	42.3 ± 6.1
II	65.9 ± 10.8	41.7 ± 8.4
III	65.5 ± 13.4	44.1 ± 8.9

*p* > 0.05, Kruskal–Wallis.

**Table 2 diagnostics-13-02405-t002:** Posselt’s envelope frontal plane, mandibular border movements.

Skeletal Class	Area (Mean ± SD, mm^2^)	Trajectory (Mean ± SD, mm)			
		Right Lateral	Left Lateral	Right Opening	Left Opening
I	426.9 ± 121.0	35.0 ± 17.0	34.6 ± 14.2	57.5 ± 17.7	59.4 ± 16.7
II	424.5 ± 139.4	26.2 ± 8.4	25.4 ± 9.2	57.7 ± 6.7	57.6 ± 8.5
III	426.5 ± 179.4	27.9 ± 12.1	23.9 ± 6.6	62.5 ± 12.4	62.8 ± 14.3

*p* < 0.05.

**Table 3 diagnostics-13-02405-t003:** Posselt’s envelope sagittal plane, mandibular border movements.

Skeletal Class	Area (Mean ± SD, mm^2^)	Trajectory (Mean ± SD, mm)		Ranges (Mean ± SD, mm)
		Retrusion and Posterior Opening	Protrusion and Front Opening	Protrusion
I	235.7 ± 86.9	78.7 ± 11.8	91.8 ± 19.0	11.1 ± 5.9
II	291.1 ± 120.8	68.6 ± 13.1	81.7 ± 11.7	11.1 ± 9.3
III	208.1 ± 147.8	73.0 ± 11.6	86.8 ± 15.7	7.7 ± 2.7

*p* < 0.05.

**Table 4 diagnostics-13-02405-t004:** Posselt’s envelope horizontal plane, mandibular border movements.

Skeletal Class	Area (Mean ± SD, mm^2^)	Trajectory (Mean ± SD, mm)		Lateral Range (Mean ± SD, mm)	
		Right Lateral and Protrusion	Left Lateral and Protrusion	Right	Left
I	107.7 ± 43.3 ^a,b^	51.6 ± 9.3	45.5 ± 8.1	6.9 ± 3.3 ^x,y^	6.7 ± 3.4
II	93.4 ± 28.9 ^a^	45.6 ± 13.8	44.9 ± 17.0	9.2 ± 1.5 ^x^	9.3 ± 1.6
III	58.8 ± 37.4 ^b^	39.5 ± 11.7	38.8 ± 12.0	6.4 ± 3.0 ^y^	5.7 ± 3.2

Different letters indicate significant differences between groups (*p* < 0.05) (Kruskal–Wallis test adjusted by the Bonferroni correction).

**Table 5 diagnostics-13-02405-t005:** Masticatory movements.

Skeletal Class	Cycle Area (Mean ± SD, mm^2^)			Speed (Mean ± SD, mm/s)		Masticatory Frequency (Mean ± SD, Cycles/Minutes)
	Frontal Plane	Sagittal Plane	Horizontal Plane	Opening	Closing	
I	43.8 ± 13.7	11.1 ± 4.8	10.9 ± 8.5 ^a^	53.9 ± 7.6	53.9 ± 7.9	84.8 ± 8.9
II	37.3 ± 11.2	19.6 ± 4.8	8.1 ± 3.9 ^a^	53.7 ± 11.7	55.2 ± 10.9	89.0 ± 17.7
III	51.2 ± 13.8	12.6 ± 5.6	18.4 ± 6.4 ^b^	57.7 ± 6.8	57.2 ± 4.8	81.9 ± 9.0

Different letters indicate significant differences between groups (*p* < 0.05) (Kruskal–Wallis test adjusted by the Bonferroni correction).

## Data Availability

Restrictions apply to the availability of these data. Data was obtained from [Orthodontic patients the Universidad de La Frontera dental clinic and are available with the permission of third party.

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
