# Peer review of "Characterization of Mandibular Border Movements and Mastication in Each Skeletal Class Using 3D Electromagnetic Articulography: A Preliminary Study"

_diagnostics, 2023, doi:10.3390/diagnostics13142405_

Round 1
Reviewer 1 Report
This paper shows the characterization of the kinematics of the mandibular movement
and mastication of skeletal classes I, II and III patients. The results of the experiments presented here are of interest, but some parts of this paper should be revised.
1. What the readers want to know is the relationship between mandibular movement of frontal, sagittal, horizontal polyhedron and skeletal pattern (severity). Authors should exhibit the correlation between them (may be using ANB angle).
2. Why is only the right lateral range of MBM parameters in class III lower? Authors should discuss more about that.
Author Response
"Please see the attachment."

Reviewer 2 Report
Characterization of Mandibular Border Movements and Mastication in each skeletal class using 3D electromagnetic articulography: A preliminary study
The article hitted an important point in Orthodontic profession.
The aim of the study is clear.
The title of the article is appropriately selected and denotes the study performed.
The references are recent and relevant in addition to this are well written and arranged and highly related to the study.
The authors reviewed well what is already written about the subject.
The methodology is clear enough.
The results are well written.
The discussion section is very well written and elaborated good with other researchers results.
Comments:
1. An important point that was worthy to be investigated or at least defended and elaborated which is the differentiation of the two divisions of Class II cases into Div. 1 and Div. 2, I guess there is kinematic and masticatory differences between the two divisions. You may add this point to the study limitations.
2. Gender differences, if any.
3. The sample size as the authors pointed out in the study limitations is small for such research
4. Inclusion and exclusion criteria may be clarified more.
5. Final language editing is needed, there are some language mistakes. Examples:
- In the abstract line 26, this statement needs to be reviewed.
- Line 104 also this statement needs to be reviewed.
Final language editing is needed, there are some language mistakes. Examples:
- In the abstract line 26, this statement needs to be reviewed.
- Line 104 also this statement needs to be reviewed.
Author Response
"Please see the attachment."

Reviewer 3 Report
|
|
|
I cannot understand the clinical significance of this study. I have a some concerns about the methodology of this study including the limited number of subjects. How many patients were checked for eligibility? It is not clear as to who selected the radiographs? |
Overall, the paper may need language help from native English speakers for better understanding
the paper may need language help from native English speakers for better understanding.
Author Response
"Please see the attachment."

Reviewer 4 Report
The comments & suggestions are highlighted in the attached manuscript

The comments & suggestions are highlighted in the attached manuscript
Author Response
"Please see the attachment."

Reviewer 5 Report
There is need to establish normative data regarding differences in movements of mandible and chewing patterns in various types of malocclusions. The authors need to be complimented for trying to fill this gap in knowledge. The manuscript is written well and study has been conducted in a good way.
The authors have highlighted differences of various movements between Skeletal class I and class III malocclusions. There are also significant differences of various movements between class I & Class II and class II & Class III. These parameters also need to be highlighted and discussed.
The manuscript is suitable for publication after adding suggested modifications.
Round 2
Reviewer 1 Report
What the readers want to know is the relationship between mandibular movement of frontal, sagittal, horizontal polyhedron and skeletal pattern (severity). Authors should exhibit the correlation between them (may be using ANB angle). But if the editors accept this article, I will follow the dicision.
The authors should make minor editing of English language.
Reviewer 3 Report
The authors have addressed my concerns.
